# FineMedLM-o1: Enhancing Medical Knowledge Reasoning Ability of LLM from Supervised Fine-Tuning to Test-Time Training

**Hongzhou Yu**[1], **Tianhao Cheng**[1], **Yingwen Wang**[2], **Wen He**[2], **Qing Wang**[2],
**Ying Cheng**[1†], **Yuejie Zhang**[1], **Rui Feng**[1,2,3†], **Xiaobo Zhang**[2†]
[1]College of Computer Science and Artificial Intelligence, Shanghai Key Laboratory of Intelligent Information Processing, Fudan University
[2]Children's Hospital of Fudan University, National Children's Medical Center, Shanghai
[3]College of Intelligent Robotics and Advanced Manufacturing, Fudan University

## Abstract

Recent advancements in large language models (LLMs) have shown promise in medical applications such as disease diagnosis and treatment planning. However, most existing medical LLMs struggle with the deep reasoning required for complex medical problems, such as differential diagnosis and medication recommendations. We propose FineMedLM-o1, which leverages high-quality medical synthetic data and long-form reasoning data for Supervised Fine-Tuning (SFT) and Direct Preference Optimization (DPO), enabling advanced dialogue and deep reasoning capabilities. Additionally, we introduce Test-Time Training (TTT) in the medical domain for the first time, facilitating domain adaptation and ensuring reliable, accurate reasoning. Experimental results demonstrate that FineMedLM-o1 achieves a 23% average performance improvement over prior models on key medical benchmarks. Furthermore, the introduction of TTT provides an additional 14% performance boost, highlighting its effectiveness in enhancing medical reasoning capabilities. To support this process, we also propose a novel method for synthesizing medical dialogue. Compared to other open-source datasets, our dataset stands out as superior in both quality and complexity. The project and data will be released on GitHub[1].

## 1 Introduction

Medical services are essential benefits that should be accessible to all individuals worldwide, as they contribute to social development and enhance people's overall satisfaction (Tian et al., 2024b; Zhang et al., 2024a; Ji et al., 2025). Recently, LLMs have made significant strides, with several closed-source general LLMs (Achiam et al., 2023; DeepSeek-AI et al., 2024) achieving impressive performance in medical applications. This progress has inspired the research community to explore the development of more advanced open-source medical LLMs. Despite these advancements, current medical LLMs still struggle to solve complex problems through deep reasoning. This limitation stems partly from suboptimal training strategies, but more critically, from the inadequacies of existing medical datasets (Allen-Zhu & Li, 2024a;c). These datasets often lack robust logical structures and fail to include essential components such as chain-of-thought (CoT) data or o1-style long-form reasoning data, both of which are crucial for teaching models how to think critically and reason effectively (Allen-Zhu & Li, 2024b). In addition, many complex medical problems cannot be resolved with direct answers alone. Instead, they require comprehensive reasoning to arrive at reliable conclusions. Figure 1 presents a comparison of the two approaches to solving complex medical problems. As illustrated, responses generated through reasoning are typically

---

† Correspondence: {chengy18,fengrui}@fudan.edu.cn, zhangxiaobo0307@163.com

[1]https://github.com/hongzhouyu/FineMed

Question: What are the key molecular and genetic mechanisms that underlie the co-evolution of host and microbial genomes, and how do these interactions contribute to the development of novel phenotypic adaptations in humans?

Qwen2.5-72B-Instruct: The co-evolution of host and microbial genomes is underpinned by a complex interplay of molecular and genetic mechanisms that facilitate mutual adaptation and interaction. Key mechanisms include the regulation of host immune responses and metabolic pathways. Host genes that control the immune system, such as those encoding pattern recognition receptors (PRRs) and cytokines, play a crucial role in recognizing and responding to microbial components. ... For instance, shifts in the microbial community composition have been linked to chronic conditions such as diabetes, inflammatory bowel disease, and obesity. ...

QwQ-32B-Preview: **Thought**...**Summarization**\n\nThe co-evolution of host and microbial genomes is a dynamic process governed by intricate molecular and genetic mechanisms. These mechanisms involve specific genes and pathways in both the host and the microbiome that facilitate their interactions, particularly through the immune system and metabolic pathways. Host genetic variation influences these interactions, with different genetic backgrounds leading to variations in microbiome composition and function. Environmental factors also play a significant role in shaping the microbiome.\n\nFrom an evolutionary standpoint, signatures of co-evolution can be identified in both human and microbial genomes, indicating adaptive changes in response to each other's pressures. ...\n\nMoreover, these interactions significantly impact human health, with alterations in the microbiome associated with various complex diseases. By elucidating the roles of host genetics, microbial communities, and environmental factors, we can gain insights into disease susceptibility and develop more effective strategies for prevention and treatment....

Figure 1: An example from our dataset illustrating the comprehensive reasoning required to provide a reliable answer. The red and green highlights indicate points where both models addressed the same aspects. However, QwQ's response is clearer and more specific. The yellow highlights represent additional content uniquely provided in QwQ's answer.

clearer, more specific, and better suited to fully addressing the given problem. Without the capacity for thoughtful reasoning, current LLMs are prone to generating incorrect responses, increasing the risk of severe medical errors and potentially leading to critical consequences (Tian et al., 2024a).

Recently, significant efforts have been devoted to enhancing the ability of LLMs in the medical domain to generate reliable and accurate responses. These approaches typically involve using powerful LLMs to rewrite datasets, thereby filling in gaps in the logical content of the corpus, or incorporating prompts to guide the model in step-by-step reasoning (Abdin et al., 2024; Xu et al., 2023; Wu et al., 2024). However, these methods fail to address the core issue that medical LLMs are unable to perform deep reasoning. This limitation persists because existing strategies neither validate the quality and complexity of synthetic data nor integrate o1-style long-form reasoning data into the training process. As a result, the generated responses may still lack the depth and rigor required for high-stakes medical decision-making. Addressing this challenge requires a more systematic approach that not only refines data quality but also fosters advanced reasoning capabilities within LLMs.

In this paper, we propose FineMedLM-o1 and a novel synthetic data generation method aimed at enhancing both the reasoning capabilities and domain adaptation of LLMs in medical contexts. FineMedLM-o1 is trained through SFT and reinforcement learning on the base model, while incorporating TTT during inference to further enhance its reasoning capabilities. The synthetic data generation pipeline is a comprehensive system encompassing instruction generation, scoring, filtering, and response generation, ensuring the creation of high-quality, domain-specific datasets that effectively support model training.

Specifically, the training process of FineMedLM-o1 consists of two steps. First, the model undergoes a fine-grained 3-stage SFT using synthetic medical dialogues. To support this training, we develop a high-quality medical SFT dataset called FineMed, generated using the synthetic data method described above. FineMed is of superior quality, having undergone quality scoring and filtering by the LLM-as-a-judge framework during its development. It consists of about 300,000 samples, which are divided into fine-grained subsets to facilitate the 3-stage SFT process. Subsequently, the model's reasoning capabilities are enhanced through DPO (Rafailov et al., 2024). This step involves further fine-tuning with medical data containing complex instructions and o1-style responses, as well as preference learning using common and o1-style responses. In this stage, we utilize 33,000 high-quality DPO pairs to refine the model's performance. To enhance reasoning capabilities, we introduce TTT, which enables the model to retrieve and learn from similar data before generating responses. This technique allows the model to better adapt to domain-specific knowledge and reasoning processes, thereby improving the robustness and reliability of its outputs. Our synthetic data generation method incorporates a robust verification framework to evaluate data quality, complexity, medical relevance, and specificity. To the best of our knowledge, we are the first to apply o1-style data and TTT in the medical domain to enhance reasoning capabilities and the first to introduce a validation method for synthetic medical data.

The main contributions are as follows: (1) We introduce a novel framework for generating large-scale, high-quality synthetic SFT data, the first of its kind for medical data, ensuring strict adherence to content, context, quality, and complexity standards. (2) We implement a complete process, from SFT and DPO to TTT, for the medical LLM FineMedLM-o1, advancing the exploration of LLM reasoning capabilities on complex medical tasks. (3) We will open-source all the code, datasets, and resources used in this research, with the goal of supporting further research and fostering innovation within the open-source community.

## 2 FineMed

### 2.1 Data Synthesis

To construct FineMed, the first step is to generate large amounts of high-quality synthetic medical data. In the following, we describe our method, as illustrated in Figure 2.

**Raw Medical Texts** Unlike approaches that rely on conversation data from real-world application scenarios on online treatment platforms or filter open-source medical SFT datasets (Yang et al., 2024b; Gururajan et al., 2024), we aim to use internet corpora (e.g., Common Crawl, CC) as the foundation for our medical knowledge texts. CC inherently includes large-scale question-answer pairs and knowledge-rich textbooks (Shao et al., 2024; Yue et al., 2024). Moreover, advancements in the classification of internet corpora by discipline, such as the open-sourced FineFineWeb (M-A-P, 2024), have significantly accelerated the development of LLMs across various domains. Leveraging this resource, we randomly selected 420,000 samples from the medical subset of FineFineWeb as our raw medical texts.

**Instruction Generation** Building on the perspective proposed by previous researchers (Lu et al., 2023), we contend that the complexity of a SFT dataset is primarily determined by the intricacy of its instructions. Concurrently, several studies (Zhou et al., 2024; Cao et al., 2023) have demonstrated that applying quality filtering to synthetic data generated by strong base models can significantly enhance performance by removing low-quality instructions. Based on these insights and with guidance from professional doctors, we establish a set of scoring criteria for generated instructions and utilize Qwen (Yang et al., 2024a) to evaluate them accordingly. It is noteworthy that we utilize vLLM (Kwon et al., 2023) to accelerate the inference process during the generation of synthetic data. Specifically, as shown in Figure 2a, we employ Qwen to generate two distinct instructions for each medical text and assign a quality and complexity score to each instruction on a scale of 1 to 10, based on predefined criteria (detailed in Appendix B.1 and B.2). To ensure the instructions remain relevant to medicine and do not excessively dilute the total quality and complexity scores, relevance to medicine is scored on a scale of 1 to 6. Furthermore, we use Qwen to evaluate whether the generated instructions include specific details from the medical text to prevent unanswerable instructions due to missing contextual information. Finally, we filter the scored instructions through a multi-step process, as outlined in Algorithm 1 (see Appendix C). This approach yields 333,000 high-quality, high-complexity instruction samples.

**Response Generation** As illustrated in Figure 1, directly answering high-complexity instructions often leads to fragile responses. To ensure the quality of responses in the SFT dataset, we first categorize instructions into common and complex types based on their complexity scores, with a threshold set at 8. The response generation process is shown in Figure 2b. For common instructions, we use Qwen to generate two stylistically distinct responses and employ a reward model[2] to select the response that better aligns with human preferences. We then validate the selected responses through a multi-stage process. First, we prompt a LLM to assess whether each response (1) appropriately addresses the given instruction and (2) is grounded in the original source text. Responses that fail either criterion are subsequently reviewed and revised by professional clinicians. Through this pipeline, we curate a high-quality SFT dataset comprising 300,000 instruction–response pairs. For

---

[2]https://huggingface.co/sfairXC/FsfairX-LLaMA3-RM-v0.1(Xiong et al., 2024)

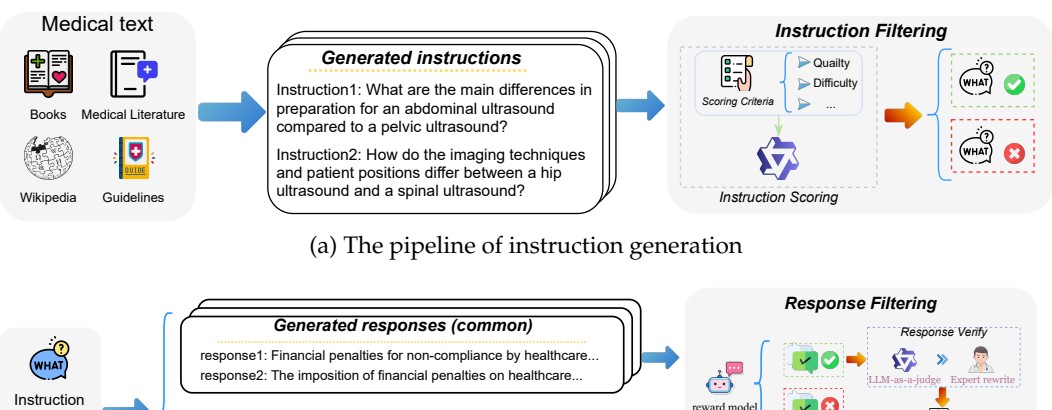

(a) The pipeline of instruction generation

(b) The pipeline of response generation

Figure 2: The overall pipeline of generating synthetic data. Figure 2a illustrates the generation of instructions derived from medical texts. Figure 2b depicts the subsequent process of generating common responses and long-form reasoning responses.

complex instructions, we engage QwQ (Team, 2024) to generate and verify detailed long-form reasoning responses, creating an SFT dataset enriched with o1-style data. Additionally, we use FineMedLM (our SFT-trained model) to generate responses to the same instructions and pair them with QwQ's reasoning outputs, forming a dataset for DPO. Detailed prompts for response generation can be found in Appendix B.3.

## 2.2 Data classification

In fine-tuning the base model for medical dialogue applications, we adopt a 3-stage SFT strategy, beginning with a broad medical domain and progressively narrowing the focus to more specific subfields. To achieve this, a fine-grained classification of medical data is essential. We introduce a medical knowledge classification diagram (see Appendix D), which integrates the department structure of Zhongshan Hospital, affiliated with Fudan University[3], alongside relevant data from the Chinese Hospital Association[4] (CHA), with additional consultation from professional doctors. Previous studies have shown that providing appropriate prompts enables LLMs to effectively classify data (Dong et al., 2024). Therefore, we employ Qwen to categorize the medical data according to the aforementioned classification framework, ultimately producing FineMed. The classification prompt used is detailed in Appendix E. FineMed is a SFT dataset comprising five primary and twenty-nine secondary categories. Key statistics of the dataset are summarized in Table 17 (see Appendix F for details). In Section 2.3, we compare the quality and complexity of FineMed with other widely used open-source medical SFT datasets and demonstrate the distribution of data across departments in the semantic space, highlighting the effectiveness of our prompts used for the classification approach.

## 2.3 Dataset Analysis

**Comparison with other datasets** To better demonstrate the advantages of our proposed synthetic data approach, we employ the LLM-as-a-judge approach to evaluate the quality and complexity of both FineMed and other medical SFT datasets. As illustrated in Figure 3,

---

[3]https://www.zs-hospital.sh.cn/
[4]https://www.cha.org.cn/

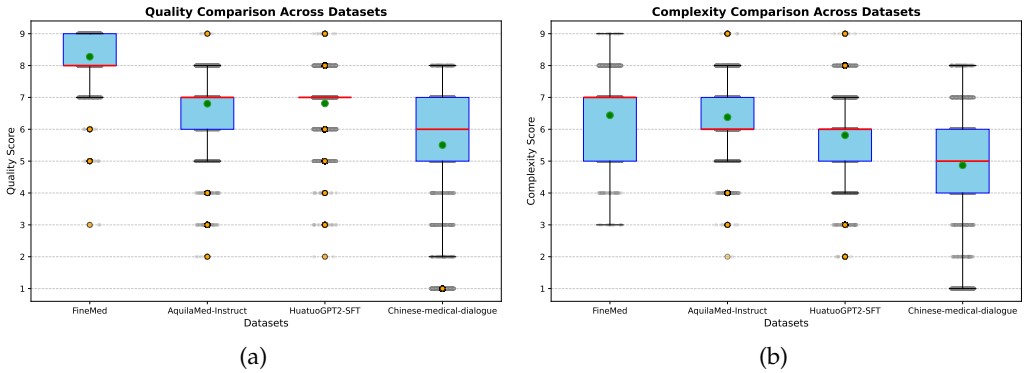

Figure 3: Comparison of four datasets in terms of quality and complexity.

we randomly select 5,000 samples each from FineMed, AquilaMed-Instruct (Zhao et al., 2024), HuatuoGPT2-SFT (Chen et al., 2023) and Chinese-med-dialogue[5] for comparison. In terms of instruction quality, FineMed achieves the highest average quality and median scores, with a relatively concentrated quality distribution. Regarding instruction complexity, both FineMed and AquilaMed-Instruct, which employs the Deita (Liu et al., 2023b) method for complexity filtering, demonstrate higher levels of complexity. This result indicates that our scoring criteria effectively capture the impact of the Deita method, with both datasets exhibiting comparable performance in this aspect.

**Verifying the robustness of classification approach**  To facilitate the analysis of data distribution across different first-level departments within FineMed, we employ a two-dimensional semantic space for visualization. This approach enables a more intuitive examination of the relationships among departmental data points, thereby providing empirical support for the validity of our classification methodology and the proposed diagram. Specifically, we leverage the t-distributed Stochastic Neighbor Embedding (t-SNE) technique to project the embeddings of FineMed's first-level department data, extracted using the bge-large-en-v1.5 model (Xiao et al., 2023), into a lower-dimensional space. The resulting visualization, depicted in Figure 4, reveals distinct separations among data clusters, underscoring the robustness and discriminative power of our classification framework.

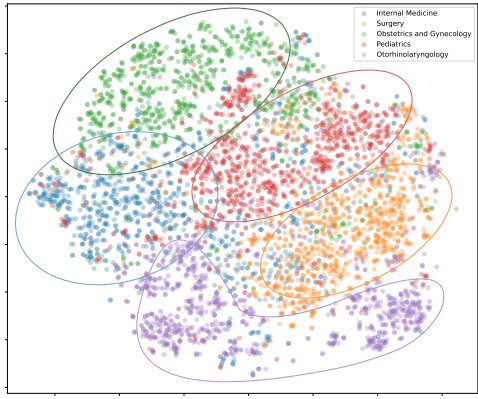

Figure 4: Visualization on the t-SNE data distributions of FineMed.

## 3   Training and Inference

In this section, we detail the two primary stages of model training: SFT and DPO. To further enhance the reasoning capabilities of the medical LLM, we also incorporate TTT during inference. The overall workflow is illustrated in Figure 5.

---

[5]https://huggingface.co/datasets/ticoAg/Chinese-medical-dialogue

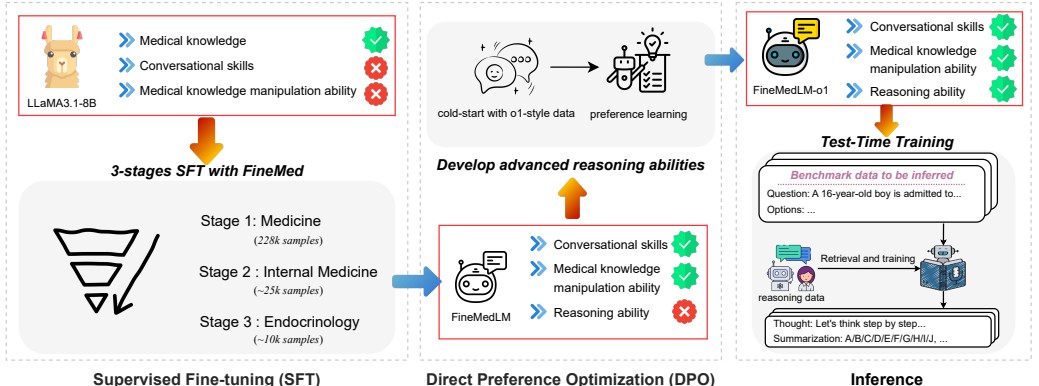

Figure 5: The overall pipeline for training and inference of FineMedLM-o1.

## 3.1 FineMedLM

To enhance the language model's capability for engaging in natural conversations, we first conduct 3-stage SFT, which involves fine-tuning the pre-trained LLM on chat-style data, including both instructions and responses.

Many researchers (Wang et al., 2021; Zhang et al., 2024b) have found that models perform better when they are trained incrementally from simple tasks to more complex ones. Based on this principle, multi-stage learning methods have gained increasing popularity. For instance, some researchers employ a multi-stage training approach for continuous pre-training (Zhao et al., 2024). In their method, the first stage involves mixed training with general and medical data, while the second stage focuses solely on medical data. This approach aims to mitigate the significant performance degradation caused by discrepancies between pre-training and fine-tuning datasets, thereby enhancing the model's capabilities in the medical domain. Inspired by this, we adopt a 3-stage SFT approach within medical knowledge, starting broadly with general medical domain data and progressively narrowing the scope to focus on increasingly specialized subfields.

Our model is based on Llama3.1-8B and fine-tuned in 3 stages. In the first stage, we randomly sample 228,000 instances from the entire medical dataset for training, using a learning rate of 1e-5. We reserve 10% of the training set for validation and obtain the best checkpoint after two epochs. In the second stage, 25,600 instances are randomly selected from the Internal Medicine subset of FineMed, with a learning rate of 5e-6, resulting in the best checkpoint after one epoch. In the third stage, 10,240 instances are selected from the Endocrinology subset, using a learning rate of 5e-6, and the best checkpoint is obtained after one epoch. The training hyperparameters are as follows: sequence length is set to 1024, batch size to 256, and warm-up steps to 10% of the total steps. A cosine learning rate scheduler is employed. To mitigate overfitting, we apply a weight decay of 0.01 and set dropout to 0.1. The training stage is parallelized across 4 NVIDIA A100-80G GPUs, utilizing the AdamW optimizer with bf16 precision and the ZeRO-3 optimization strategy.

## 3.2 FineMedLM-o1

We obtain FineMedLM by conducting a 3-stage SFT on Llama3.1-8B using carefully curated non-reasoning data. It is a model equipped with both dialogue abilities and medical knowledge but lacks deep reasoning capabilities. To develop FineMedLM-o1 with advanced reasoning capabilities, we apply DPO to FineMedLM. We first utilize dialogue data containing explicit reasoning traces to cold-start the model, thereby equipping it with preliminary reasoning capabilities. Subsequently, for each complex instruction, we perform preference learning using both the verified correct responses (which include reasoning processes) generated by QwQ and the verified incorrect responses generated by the cold-started model, in order to further enhance its ability to reason effectively.

During the cold-start stage, we randomly select 12,800 samples from the FineMed subset for training, which includes complex instructions and long-form reasoning responses. The hyperparameters are set as follows: a learning rate of 5e-6, a batch size of 128, a sequence length of 8192, and 2 epochs. All other hyperparameters are kept consistent with those used for FineMedLM. In the preference learning stage, we utilize all 33,000 samples to construct the DPO dataset and train the model for one epoch with a learning rate of 1e-7, while keeping all other hyperparameters consistent with those used in the cold-start stage.

### 3.3 Test-Time Training

In order to further advance the reasoning capabilities of FineMedLM-o1, we incorporate TTT into the inference process. Some researchers demonstrate that TTT can substantially enhance a model's reasoning abilities (Akyürek et al., 2024; Hübotter et al., 2025). Specifically, their findings reveal that incorporating TTT can enhance performance by six times on the Abstraction and Reasoning Corpus (ARC) (Chollet, 2019), a challenging benchmark for assessing reasoning abilities, compared to a baseline fine-tuned model.

When making inferences on a benchmark dataset, we first use bge-large-en-v1.5 to retrieve the most similar instance from the long-form reasoning subset of FineMed. The model then undergoes training on the retrieved data using the same hyperparameters as FineMedLM-o1. Once training is completed, the model generates an answer for the benchmark instance, after which the model parameters are restored to their original state. This approach ensures that the model leverages relevant contextual knowledge while maintaining its generalization ability. By reverting the parameters, we prevent catastrophic forgetting and preserve the model's original capabilities for subsequent tasks.

## 4 Experiment

### 4.1 Evaluation Setup

We evaluate the performance of our model on several Chinese and English benchmarks in the medical field. These benchmarks span multiple domains, including biology and healthcare, and encompass both common medical questions and challenging problems that demand complex reasoning. This comprehensive evaluation assesses the model's ability to understand medical knowledge and provide accurate answers to medical queries.

**Benchmarks**   We evaluate the models using medical question subsets from the MMLU (Hendrycks et al., 2021) and C-Eval (Huang et al., 2024b) benchmarks, along with questions from the CMB-Exam (Wang et al., 2023), CMExam (Liu et al., 2023a), MedQA (Jin et al., 2021), and MedMCQA (Pal et al., 2022) test sets to assess their proficiency in medical knowledge. To more clearly illustrate the impact of long-form reasoning data on enhancing the model's reasoning capabilities through DPO and TTT, we incorporate additional medical subsets from the MMLU-Pro benchmark (Wang et al., 2024b) for evaluation. These subsets feature more challenging tasks that require complex reasoning, thereby providing a robust assessment of the model's advanced reasoning ability. Among these benchmarks, C-Eval, CMB-Exam, and CMExam consist of Chinese questions, while the others are in English.

**Baselines**   For common medical benchmarks, we compare the performance of our models against both general models and medical fine-tuned models with comparable parameter sizes. The models in our comparison include Baichuan2-7B (Baichuan, 2023), ChatGLM3-6B (GLM et al., 2024), InternLM-7B (Cai et al., 2024), Llama3.1-8B (Dubey et al., 2024), HuatuoGPT2-7B (Chen et al., 2023), and Medical-Llama3-8B[6]. For benchmarks that require complex reasoning, we compare with the medical reasoning model HuatuoGPT-o1-8B (Chen et al., 2024). Additionally, we report experimental results for QwQ-32B-Preview (Team, 2024), GPT-4o-mini (OpenAI, 2024), GPT-4o, and the recently released DeepSeek-v3 (DeepSeek-AI et al., 2024) as supplementary references.

---

[6]https://huggingface.co/ruslanmv/Medical-Llama3-8B

| Model | C-Eval | CMB-Exam | CMExam | MMLU | MedQA | MedMCQA |
|---|---|---|---|---|---|---|
| Baichuan2-7B(Baichuan, 2023) | 56.96 | 51.13 | 50.70 | 50.14 | 41.56 | 41.12 |
| ChatGLM3-6B(GLM et al., 2024) | 37.97 | 42.94 | 42.01 | 46.74 | 33.78 | 33.71 |
| InternLM2.5-7B(Cai et al., 2024) | 53.16 | 48.09 | 49.42 | 60.61 | 45.01 | 40.08 |
| HuatuoGPT2-7B ♠(Chen et al., 2023) | 51.90 | 45.79 | 45.29 | 45.64 | 40.38 | 34.93 |
| Medical-Llama3-8B ♠ | 40.51 | 35.09 | 34.87 | 65.47 | 50.20 | 50.68 |
| Llama3.1-8B(Dubey et al., 2024) | 43.04 | 40.80 | 41.83 | 67.03 | 55.30 | 53.77 |
| FineMedLM ♠ | 55.70 | 48.47 | 46.32 | 71.44 | 57.66 | 54.91 |
| FineMedLM-o1 ♠ | 62.03 | 54.74 | 51.93 | 78.24 | 58.52 | 62.75 |
| FineMedLM-o1 (TTT) ♠ | 65.75 | 58.02 | 55.04 | 81.36 | 60.86 | 65.26 |

Table 1: Main results on medical benchmarks. ♠ means this LLM is specifically fine-tuned and optimized for tasks within the medical domain. Within each segment, **bold** highlights the best scores, and underlines indicate the second-best.

| Model | Size | Biology | Health | Average |
|---|---|---|---|---|
| FineMedLM | 8B | 58.72 | 42.42 | 50.57 |
| HuatuoGPT-o1(Huang et al., 2024a) | 8B | 68.20 | 58.70 | 63.45 |
| FineMedLM-o1 | 8B | 70.71 | 57.95 | 64.33 |
| FineMedLM-o1 (TTT) | 8B | 80.54 | 66.71 | 72.83 |
| GPT-4o-mini(OpenAI, 2024) | - | 80.20 | 67.60 | 73.90 |
| QwQ-32B-Preview(Team, 2024) | 32B | 84.10 | 70.66 | 77.38 |
| GPT-4o(OpenAI, 2024) | - | 86.75 | 72.12 | 79.44 |
| DeepSeek-v3(DeepSeek-AI et al., 2024) | 671B | 88.15 | 74.82 | 81.49 |

Table 2: Main results on MMLU-Pro. **bold** highlights the best scores, and underlines indicate the second-best.

## 4.2 Results

In the experiments, we randomly select three data points from the benchmark training set to perform 3-shot evaluations, ensuring consistency in the model output format. To mitigate potential bias, the experiment is repeated three times, and the average value of the results is reported as the final outcome.

Table 1 summarizes the overall performance of various models on standard medical benchmarks. Notably, some newer models (e.g., Llama3.1-8B), including general-purpose models, occasionally surpass specialized medical fine-tuned models on certain benchmarks. Our model, FineMedLM, achieves significant improvements across all benchmarks compared to its base model, Llama3.1-8B, with an average performance gain of 12%. However, FineMedLM underperforms on Chinese benchmarks (C-Eval, CMB-Exam, CMExam) relative to Baichuan2-7B, which benefits from pretraining on extensive Chinese datasets. FineMedLM-o1 exhibits strong performance across all benchmarks, outperforming FineMedLM by an average of 10%, highlighting the critical role of robust reasoning capabilities in addressing medical problems.

Table 2 presents our performance on challenging medical benchmarks that require complex reasoning. FineMedLM-o1 shows a significant improvement in reasoning ability over FineMedLM, with a gain of approximately 27%.Notably, FineMedLM-o1 achieves superior average performance compared to the recently released HuatuoGPT-o1 on the medical subset of MMLU-Pro. This phenomenon can be attributed to our carefully crafted synthetic data and optimized training process. Furthermore, the introduction of TTT further boosts FineMedLM-o1's reasoning capabilities, bringing its performance on par with GPT-4o-mini.

## 4.3 Ablation Studies

In this section, we conduct four studies: (1) evaluating the effectiveness of our 3-stage SFT process, (2) verifying the consistency between our LLM-as-a-judge framework and expert assessments of instruction quality and complexity, (3) analyzing the impact of long-form reasoning data on TTT, and (4) comparing the additional inference time introduced by TTT.

| Strategy | C-Eval | CMB-Exam | CMExam | MMLU | MedQA | MedMCQA |
|---|---|---|---|---|---|---|
| Direct | 49.37 | 48.04 | 45.69 | 70.33 | 55.54 | 54.48 |
| Reversed | 48.10 | 47.95 | 45.42 | 70.06 | 56.33 | 52.69 |
| FineMedLM | **55.70** | **48.47** | **46.32** | **71.44** | **57.66** | **54.91** |

Table 3: Results of ablation experiments for 3-stage SFT. "Direct" means training directly with medical data, and "Reversed" means training by reversing the order of the SFT stages.

**Ablation for 3-stage SFT**    To assess the effectiveness of the proposed 3-stage SFT approach, we design two baseline methods for comparison. The first baseline involves training solely on FineMed's medical dataset without incorporating multiple stages. The second baseline employs a 3-stage SFT process but with the stages reversed relative to FineMedLM. The results, summarized in Table 3, demonstrate that FineMedLM outperforms the baseline across all benchmarks, achieving a maximum performance improvement of 15%. To further demonstrate the effectiveness of the proposed 3-stage SFT framework, we conduct additional experiments on a private benchmark derived from a hospital dataset, with a focus on internal medicine and endocrinology. We evaluate models trained at each stage of the SFT process alongside several strong open-source baselines. A summary of the results is provided in Appendix H. These findings underscore the significant contribution of the 3-stage SFT framework in enhancing the model's ability to effectively encode and utilize medical knowledge.

**Expert Evaluation Comparison**    To assess the consistency of our LLM-as-a-judge approach with expert evaluations, we randomly sample 3,000 instances from the FineMed dataset and invite professional clinicians to evaluate them using the same criteria applied by the LLM. The evaluation results, along with the original LLM scores for comparison, are presented in Table 4. Notably, we observe a high degree of agreement between human and LLM assessments, supporting the reliability of our automatic evaluation method.

| Evaluator | Quality (Average Score) | Complexity (Average Score) |
|---|---|---|
| Expert | 8.35 | 6.33 |
| Qwen2.5-72B-Instruct(Yang et al., 2024a) | 8.27 | 6.41 |

Table 4: Main results on expert quality evaluation. We randomly sample 3,000 instances from the FineMed dataset and invite professional physicians to evaluate them using the same criteria as in the LLM-based assessment. For comparison, we also include the original LLM-based scores.

**Ablation for long-form reasoning data**
To evaluate the impact of long-form reasoning data on TTT, we compare its performance using common data versus reasoning data. As shown in Table 5, TTT with reasoning data out-

| Method | Biology | Health |
|---|---|---|
| No TTT | 70.71 | 57.95 |
| TTT with common data | 72.83 | 59.79 |
| TTT with reasoning data | **80.54** | **66.71** |

Table 5: Results of ablation experiments for TTT.

performs TTT with common data across all benchmarks, achieving an average improvement of 11%. These results underscore the significant contribution of long-form reasoning data in enhancing TTT's effectiveness. Moreover, comparing the performance changes after introducing TTT in Table 1 and Table 2 indicates that TTT yields even greater benefits for complex problems that require deep thinking.

**Inference Time Comparison**    To better examine the trade-off between inference efficiency and performance gains, we evaluate three models: a non-thinking conversation model, a thinking model, and a thinking model incorporating TTT. The comparison focuses on their inference time and corresponding reasoning behavior, as shown in Table 6. From FineMedLM to FineMedLM-o1, the inference time increases by approximately 4.7×, accompanied by the emergence of thinking behaviors. With the introduction of TTT, inference time further increases by 2.3×, while the model exhibits more salient reasoning behaviors. We

| Model | Inference Time | Reasoning Behavior |
|---|---|---|
| FineMedLM | 3.3s | Absent |
| FineMedLM-o1 | 15.4s | Present |
| FineMedLM-o1 with TTT | 35.1s | Enhanced |

Table 6: Comparison of the average inference time per instance and corresponding reasoning behavior among FineMedLM, FineMedLM-o1, and FineMedLM-o1 with TTT.

present a case study that qualitatively compares the outputs of the three models mentioned above. The example, drawn from the MMLU-Pro benchmark, involves a complex inference task and is detailed in Appendix G.

## 5 Related Work

The development of medical LLMs is of profound significance to human society. These models can assist doctors in making rapid and accurate diagnoses, formulating treatment plans by integrating extensive medical data and clinical cases, and even supporting medical institutions in optimizing resource allocation to enhance the efficiency and quality of healthcare services (Thirunavukarasu et al., 2023; Wang et al., 2024a). Consequently, researchers have pursued diverse approaches to building powerful medical LLMs. For instance, ChatDoctor (Li et al., 2023) utilizes patient-doctor conversation data and is fine-tuned on Llama to improve language model accuracy in healthcare applications. DISC-MedLLM (Bao et al., 2023) employs a two-stage SFT strategy: the first stage integrates medical domain knowledge and conversational capabilities into the model using large-scale instruction datasets, while the second stage fine-tunes it on a smaller, high-quality dataset curated with the assistance of ChatGPT and filtered by human experts to align with human preferences.

Despite these advancements, existing models often struggle to deliver professional responses to domain-specific medical questions. To this end, some researchers propose PediatricsGPT (Yang et al., 2024b), a model tailored to pediatrics, which leverages high-quality instruction datasets and intricate training procedures, contributing significantly to pediatric expertise in LLMs. However, these models struggle to solve complex medical problems through step-by-step reasoning and self-correction. Our work builds upon the paradigm of using medical knowledge to perform SFT and DPO on LLMs, introducing innovations in data utilization and training strategies to enhance reasoning capabilities for complex medical problems. Moreover, we address a critical gap in domain-specific medical datasets by releasing high-quality SFT datasets encompassing 5 primary medical specialties and 29 subspecialties, providing a valuable resource to the community. Notably, shortly before the publication of our study, HuatuoGPT-o1 (Chen et al., 2024) was launched. This medical reasoning model, trained using SFT and Proximal Policy Optimization (PPO) (Schulman et al., 2017), provides valuable insights for enhancing the complex medical reasoning capabilities of LLMs. We believe that our research not only advances the field of medical LLMs but also provides valuable methodologies and resources to inspire progress in enhancing LLM reasoning abilities across medicine and other specialized domains.

## 6 Conclusion

In this paper, we propose FineMedLM-o1, a LLM with strong medical reasoning capabilities, designed to address the challenges of complex medical problems by performing SFT and DPO with carefully designed synthetic data. Our novel synthetic dataset construction, training process, and introduction of TTT during inference significantly improve the model's ability to handle deep reasoning on complex medical problems. The outstanding performance of FineMedLM-o1 on various benchmarks validates the effectiveness of our approach. By open-sourcing our dataset and training process, we aim to promote the advancement of complex reasoning capabilities of medical LLMs in the research community.

## Acknowledgments

This work was supported by the National Natural Science Foundation of China (No. 62172101), and the Science and Technology Commission of Shanghai Municipality(No. 23511100602), and Shanghai Municipal Commission of Economy and Informatization, Corpus Construction for Large Language Models in Pediatric Respiratory Diseases (2024-GZL-RGZN-01013)

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

## A    Limitations

Our data generation pipeline is built on seeds from FineFineWeb, which in turn are derived from FineWeb. Both datasets have undergone multiple rounds of deduplication and quality filtering to enhance data diversity and reliability. Nevertheless, additional quality assessment may remain beneficial. Due to limited computational resources, however, we randomly selected raw medical texts during the data selection phase without performing prior quality evaluation. Additionally, the number of selected texts is relatively small. Recognizing the critical role of diverse and high-quality data in SFT, our future work aims to incorporate more extensive, higher-quality, and larger medical datasets. Furthermore, we are exploring improvements to the DPO stage. In future research, we aspire to develop a reinforcement learning algorithm specifically tailored to the medical domain, enabling more sophisticated analyses and applications in medical scenarios.

## B    Synthetic Data Generation Prompts

This section provides a comprehensive overview of all the prompts utilized during the pipeline of generating synthetic data.

### B.1    Instruction Generation Prompt

---

**INSTRUCTION GENERATION**

---

You need to generate two questions based on the given text content. These questions can be open-ended, detail-oriented, or related to a broader discussion of the content, but avoid relying on specific case details from the text. Follow these requirements:
Requirements:
1. Make sure the questions are closely related to the main points or themes mentioned in the text.
2. Ensure the two questions are as diverse as possible, avoiding homogeneity.
3. Ensure the questions include all the information needed for the answers. If necessary, add introductory information to the questions.
4. Avoid repetitive or overly simplistic questions, ensuring diversity and depth.
5. The questions must be self-contained and should not require the provided text as background to be understood.
Please rewrite the following text into related questions, and strictly follow the format below for output:
{{
    "question1": "Generated first question",
    "question2": "Generated second question"
}}
Here is the sample text:
{text}

---

Table 7: Prompts for generating instructions. The text refers to individual data samples drawn from the medical subset of FineFineWeb.

## B.2 Instruction Scoring Prompt

---

**INSTRUCTION SCORE GENERATION**

---

You need to evaluate the given query based on the following criteria and output the results in JSON format. The output should include three parts: quality, difficulty, and whether additional necessary information is required to answer the query. Please follow the scoring standards below:

1. Quality (Score 1–10): Assess the clarity and accuracy of the query. If the query is a simple statement without any question or instruction, score it 1–2.

      9–10: Very clear, accurate expression, no ambiguity.

      7–8: Clear, accurate expression, but may have minimal ambiguity.

      5–6: Fairly clear, generally accurate expression, but some ambiguity exists.

      3–4: Not very clear, somewhat vague expression, with obvious ambiguity.

      1–2: Unclear, very vague expression, difficult to understand or a simple statement.

2. Difficulty (Score 1–10): Assess the difficulty of understanding and answering the query.

      9–10: Very difficult, requires specialized knowledge and complex analysis to answer.

      7–8: Quite difficult, requires some specialized knowledge and analysis.

      5–6: Moderate difficulty, requires general knowledge and analysis.

      3–4: Fairly simple, can be answered with basic knowledge.

      1–2: Very simple, no special knowledge required to answer.

3. Relevance to medicine (1–6): Assess the medical relevance of the query.

      5–6: Completely related to medicine, with many medical terms appearing.

      3–4: Related to medicine, the content involves medical fields.

      1–2: Very weak medical relevance.

4. Mention specific details: Whether specific case details in the text are mentioned.

Please strictly follow the format below for output:

{{

    "quality": 1–10,

    "difficulty": 1–10,

    "Relevance2Medicine": 1–6,

    "MentionSpecificDetails": "True"/"False"

}}

Please evaluate the following query:

{instruction}

---

Table 8: Prompts for generating scores for all instructions. The instruction refers to a single data sample generated by Qwen.

## B.3 Response Generation Prompts

---

**QWEN RESPONSE GENERATION**

---

You need to generate two different styles of answers based on the given question. Use the background information provided in the text to assist in formulating a relevant and detailed answer. Follow these answer guidelines:
1. Ensure the answer is closely related to the main points or themes mentioned in the question.
2. Utilize the text content to provide a comprehensive and accurate answer.
3. Ensure proper formatting and readability, including the correct rendering of any LaTeX or mathematical symbols.
4. Ensure that the aMedClsnswer provides a complete solution or explanation, with clear and detailed steps.
5. Please strictly follow the format below for output:
{{
    "answer1": "Generated first answer content",
    "answer2": "Generated second answer content"
}}
Here is the question:
{instruction}
Here is the text:
{text}

---

**QWQ RESPONSE GENERATION**

---

You need to generate an answer based on the given problem and thoroughly explore the problem through a systematic and long-term thinking process to provide a final and accurate solution. This requires a comprehensive cycle of analysis, summary, exploration, re-evaluation, reflection, backtracking and iteration to form a thoughtful thinking process. Use the background information provided in the text to assist in formulating the answer. Follow these answer guidelines:
1. Please structure your response into two main sections: Thought and Summarization.
2. During the thinking phase, think step by step based on the given text content. If the text content is used, it must be expressed.
3. During the summary phase, based on the thinking process in the thinking phase, give the final answer to the question.
Here is the question:
{instruction}
Here is the text:
{text}

---

**RESPONSE VERIFICATION**

---

You need to determine whether the model's answer is correct, based on the given instruction and the original text that contains both the question (as part of the instruction) and its corresponding answer. Please first think carefully and analyze before making a final judgment.
Evaluate the model's answer from the following two perspectives:
1. Instruction Following: Does the answer directly respond to the question in the instruction? If the content of the answer is irrelevant or unrelated to the instruction, it should be considered incorrect.
2. Answer Grounding: Is the answer supported by the original text? If any part of the answer cannot be verified or retrieved from the original text, it should be considered incorrect.
If both conditions are met, the answer is correct. At the end, respond with only "Yes" or "No".
Here is the instruction:
{instruction}
Here is the original text:
{text}
Here is the model's answer:
{answer}

---

Table 9: Prompts for generating and verifying responses. Qwen generates high-quality direct responses, while QwQ generates long-form reasoning responses and performs verification.

## C Algorithm for comparing instruction scores

---

**Algorithm 1** Algorithm for comparing instruction scores

---

1: **Input**: item1: the score set of the first instruction; item2: the score set of the second instruction (The score set consists of three evaluation metrics—quality, complexity, and RelevanceToMedicine—as well as a boolean indicator MentionSpecificDetails denoting whether the response includes specific details.)
2: **Output**: item1 or item2 or None
3: **if** item1.MentionSpecificDetails **XOR** item2.MentionSpecificDetails **then**
4:    **if** item1.MentionSpecificDetails **then**
5:       **return** item2
6:    **else**
7:       **return** item1
8:    **end if**
9: **end if**
10: **if** item1.MentionSpecificDetails **AND** item2.MentionSpecificDetails **then**
11:    **return** None
12: **end if**
13: score1 ← item1.quality + item1.complexity + item1.RelevanceToMedicine
14: score2 ← item2.quality + item2.complexity + item2.RelevanceToMedicine
15: **if** score1 ≠ score2 **then**
16:    **return** item1 **if** score1 > score2 **else** item2
17: **end if**
18: score1 ← item1.quality + item1.complexity
19: score2 ← item2.quality + item2.complexity
20: **if** score1 ≠ score2 **then**
21:    **return** item1 **if** score1 > score2 **else** item2
22: **end if**
23: **if** item1.quality ≠ item2.quality **then**
24:    **return** item1 **if** item1.quality > item2.quality **else** item2
25: **end if**
26: **return** randomChoice({item1, item2})

---

## D Medical Knowledge Tree Classification Diagram

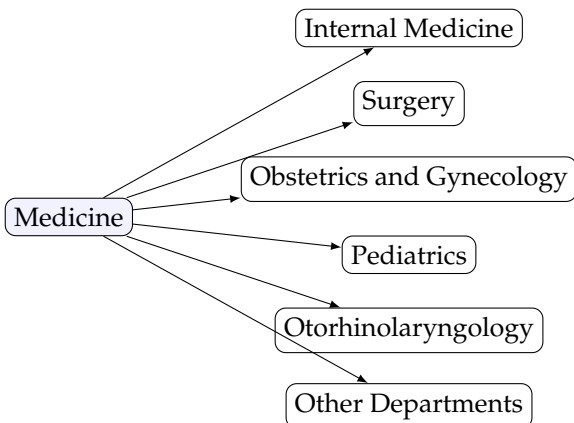

Figure 6: Hierarchical tree structure of Medicine and its subsets.

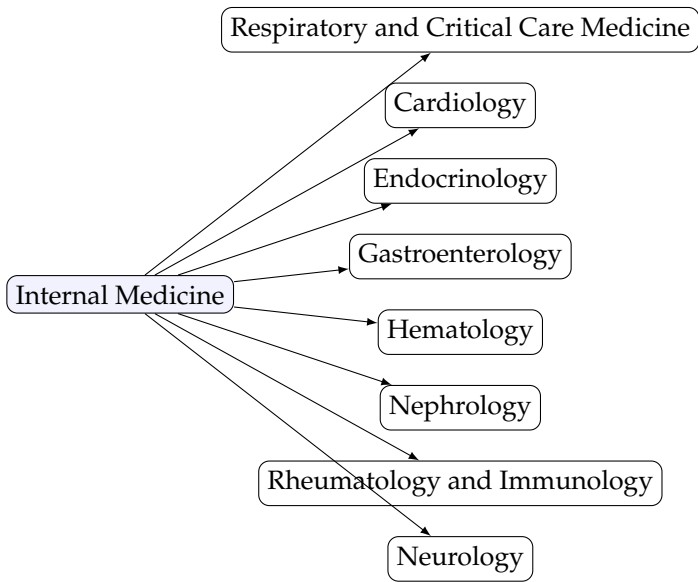

Figure 7: Hierarchical tree structure of Internal Medicine and its subsets.

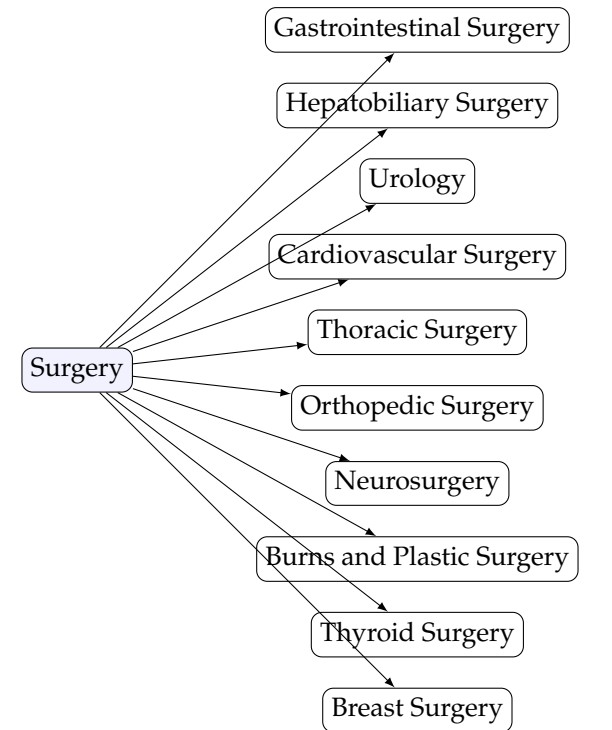

Figure 8: Hierarchical tree structure of Surgery and its subsets.

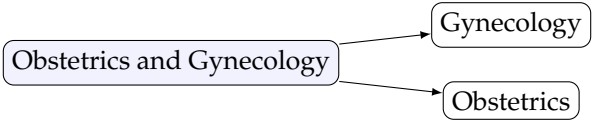

Figure 9: Hierarchical tree structure of Obstetrics and Gynecology and its subsets.

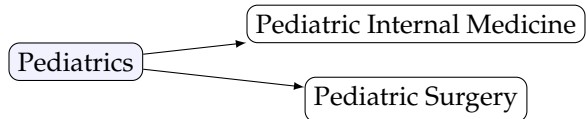

Figure 10: Hierarchical tree structure of Pediatrics and its subsets.

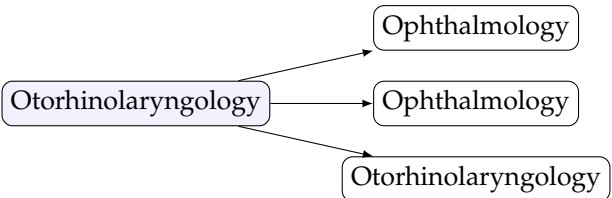

Figure 11: Hierarchical tree structure of Otorhinolaryngology and its subsets.

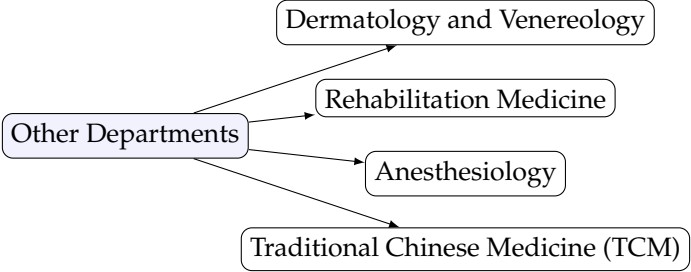

Figure 12: Hierarchical tree structure of Other Departments and its subsets.

# E   Medical Data Classification Prompts

---

**MEDICAL DATA CLASSIFICATION**

---

You are a professional doctor who can classify a dialogue. Please read the following dialogue and determine which hospital department it belongs to based on its main content. The available department categories and their brief descriptions are as follows:

Internal Medicine: Diseases related to internal organs, such as heart disease, stomach problems, diabetes, etc.

Surgery: Diseases treated with surgical interventions, such as fractures, tumor removal, etc.

Obstetrics and Gynecology: Diseases and health issues related to the female reproductive system, pregnancy, and childbirth.

Pediatrics: Health and disease issues related to children.

Otorhinolaryngology (ENT): Diseases related to the ears, nose, throat, eyes, and oral cavity.

Other Departments: Diseases or issues that do not fall under any of the above departments.

Please choose the most appropriate department from the list above and output only the department name, without any additional explanation.

Here is the dialogue:

Patient: {instruction}

Doctor: {response}

Output:

---

Table 10: Prompt for further segmentation of medical data.

---

**INTERNAL MEDICINE DATA CLASSIFICATION**

---

You are a professional doctor who can classify a dialogue. Please read the following dialogue and determine which specific department within Internal Medicine it belongs to based on its content. The available department categories and their brief descriptions are as follows:

Respiratory and Critical Care Medicine: Treats respiratory system diseases such as asthma, pneumonia, chronic obstructive pulmonary disease (COPD), and provides care for critically ill patients.

Cardiology: Covers diseases related to the heart and blood vessels, such as hypertension, coronary artery disease, arrhythmias, etc.

Endocrinology: Focuses on diseases related to endocrine glands, such as diabetes, thyroid disorders, metabolic disorders, etc.

Gastroenterology: Involves diseases of the digestive system, such as gastric ulcers, hepatitis, enteritis, etc.

Hematology: Addresses blood-related diseases, such as anemia, leukemia, lymphoma, etc.

Nephrology: Involves kidney diseases, such as nephritis, renal failure, uremia, etc.

Rheumatology and Immunology: Deals with rheumatic diseases and immune system disorders, such as rheumatoid arthritis, systemic lupus erythematous, etc.

Neurology: Focuses on diseases of the nervous system, such as stroke, epilepsy, Parkinson's disease, etc.

If the dialogue content does not pertain to any of the above departments, please output 'None'.

Please choose the most appropriate department from the list above and output only the department name, without any additional explanation.

Here is the dialogue:

Patient: {instruction}

Doctor: {response}

Output:

---

Table 11: Prompt for further segmentation of internal medicine data.

**SURGICAL DATA CLASSIFICATION**

You are a professional doctor who can classify a dialogue. Please read the following dialogue and determine which specific department within Surgery it belongs to based on its content. The available department categories and their brief descriptions are as follows:

Gastrointestinal Surgery: Mainly involves surgical issues of the digestive system, such as stomach cancer, intestinal obstruction, etc.

Hepatobiliary Surgery: Primarily involves surgical issues related to the liver, gallbladder, and pancreas, such as liver cancer, cholecystitis, pancreatic tumors, etc.

Urology: Involves surgical issues of the urinary system, such as kidney stones, bladder tumors, benign prostatic hyperplasia, etc.

Cardiovascular Surgery: Focuses on surgical diseases of the heart and major blood vessels, such as heart valve disease, aortic aneurysm, etc.

Thoracic Surgery: Covers surgical procedures involving chest organs (lungs, esophagus, etc.), such as lung cancer and esophageal cancer.

Orthopedic Surgery: Involves surgical diseases of bones, joints, and related structures, such as fractures, arthritis, herniated discs, etc.

Neurosurgery: Handles surgical issues of the central and peripheral nervous systems, such as brain tumors, cerebral hemorrhage, spinal stenosis, etc.

Burns and Plastic Surgery: Primarily responsible for burn treatment and plastic surgery procedures, such as burn scar repair and facial reconstruction, etc.

Thyroid Surgery: Deals with surgical diseases of the thyroid and related glands, such as thyroid nodules and thyroid tumors, etc.

Breast Surgery: Involves surgical treatment of breast diseases, such as breast hyperplasia and breast cancer.

If the dialogue content does not pertain to any of the above departments, please output 'None'.

Please choose the most appropriate department from the list above and output only the department name, without any additional explanation.

Here is the dialogue:

Patient: {instruction}

Doctor: {response}

Output:

Table 12: Prompt for further segmentation of surgical data.

**OBSTETRICS AND GYNECOLOGY DATA CLASSIFICATION**

You are a professional doctor who can classify a dialogue. Please read the following dialogue and determine which sub-department within Obstetrics and Gynecology it belongs to based on its content. The available sub-departments and their brief descriptions are as follows:

Gynecology: Covers diseases and health issues related to the female reproductive system, such as menstrual disorders, infertility, uterine diseases, etc.

Obstetrics: Involves pregnancy, childbirth, and postpartum care, such as pregnancy health management, delivery processes, and postpartum recovery.

If the dialogue content does not pertain to any of the above sub-departments, please output 'None'.

Please choose the most appropriate department from the list above and output only the department name, without any additional explanation.

Here is the dialogue:

Patient: {instruction}

Doctor: {response}

Output:

Table 13: Prompt for further segmentation of obstetrics and gynecology data.

**PEDIATRIC DATA CLASSIFICATION**

You are a professional doctor who can classify a dialogue. Please read the following dialogue and determine which specific department within Pediatrics it belongs to based on its content. The available sub-departments and their brief descriptions are as follows:

Pediatric Internal Medicine: Involves the diagnosis and treatment of internal diseases in children, including respiratory diseases, digestive system diseases, endocrine disorders, etc., such as colds, asthma, gastrointestinal issues, etc.

Pediatric Surgery: Involves the diagnosis and treatment of surgical diseases in children, including trauma, congenital deformities, urinary system issues, etc., such as fractures, hernias, etc.

If the dialogue content does not pertain to any of the above sub-departments, please output 'None'.

Please choose the most appropriate department from the list above and output only the department name, without any additional explanation.

Here is the dialogue:

Patient: {instruction}

Doctor: {response}

Output:

Table 14: Prompt for further segmentation of pediatric data.

**OTORHINOLARYNGOLOGY DATA CLASSIFICATION**

You are a professional doctor who can classify a dialogue. Please read the following dialogue and determine which specific department within Otorhinolaryngology (ENT) it belongs to based on its content. The available sub-departments and their brief descriptions are as follows:

Otorhinolaryngology (ENT): Involves diseases and issues related to the ears, nose, and throat, such as tinnitus, nasal congestion, sore throat, etc.

Ophthalmology: Focuses on eye health issues, such as blurred vision, eye pain, dry eye, etc.

Dentistry (Oral Medicine): Deals with issues related to the mouth, teeth, gums, etc., such as toothaches, mouth ulcers, gum bleeding, etc.

If the dialogue content does not pertain to any of the above sub-departments, please output 'None'.

Please choose the most appropriate department from the list above and output only the department name, without any additional explanation.

Here is the dialogue:

Patient: {instruction}

Doctor: {response}

Output:

Table 15: Prompt for further division of ENT data.

---

**OTHER DEPARTMENT DATA CLASSIFICATION**

---

You are a professional doctor who can classify a dialogue. Please read the following dialogue and determine which hospital department it belongs to based on its main content. The available department categories and their brief descriptions are as follows:

Dermatology and Venereology: Primarily deals with skin diseases, sexually transmitted diseases, and related skin issues.

Rehabilitation Medicine: Focuses on restoring patients' physical functions, treating sports injuries, and post-surgical rehabilitation.

Anesthesiology: Mainly responsible for anesthesia management during surgeries, including general anesthesia and local anesthesia.

Traditional Chinese Medicine (TCM): Uses traditional Chinese medicine theories and methods to treat diseases, including acupuncture, herbal medicine, and massage (Tui Na).

If the dialogue content does not pertain to any of the above departments, please output 'None'.

Please choose the most appropriate department from the list above and output only the department name, without any additional explanation.

Here is the dialogue:

Patient: {instruction}

Doctor: {response}

Output:

---

Table 16: Prompt for further division of data of other departments.

# F  Statistics of the FineMed dataset

| Medical Specialty | Count | Quality | Complexity |
|---|---|---|---|
| Respiratory and Critical Care Medicine | 4766 | 8.40 | 6.42 |
| Cardiology | 10152 | 8.40 | 6.38 |
| Endocrinology | 11477 | 8.39 | 6.43 |
| Gastroenterology | 7102 | 8.34 | 6.19 |
| Hematology | 2068 | 8.47 | 6.51 |
| Nephrology | 1728 | 8.40 | 6.46 |
| Rheumatology and Immunology | 4662 | 8.40 | 6.42 |
| Neurology | 3390 | 8.51 | 6.67 |
| Gastrointestinal Surgery | 831 | 8.53 | 6.68 |
| Hepatobiliary Surgery | 422 | 8.54 | 6.57 |
| Urology | 431 | 8.53 | 6.78 |
| Cardiovascular Surgery | 444 | 8.50 | 6.75 |
| Thoracic Surgery | 349 | 8.58 | 6.81 |
| Orthopedic Surgery | 4818 | 8.34 | 6.30 |
| Neurosurgery | 1208 | 8.46 | 6.71 |
| Burns and Plastic Surgery | 2589 | 8.18 | 6.14 |
| Thyroid Surgery | 91 | 8.70 | 6.70 |
| Breast Surgery | 1372 | 8.39 | 6.48 |
| Gynecology | 8667 | 8.36 | 6.37 |
| Obstetrics | 7559 | 8.36 | 6.24 |
| Pediatric Internal Medicine | 1729 | 8.40 | 6.26 |
| Pediatric Surgery | 298 | 8.49 | 6.59 |
| Otorhinolaryngology | 4923 | 8.27 | 6.06 |
| Ophthalmology | 736 | 8.25 | 5.79 |
| Dentistry (Oral Medicine) | 2128 | 8.28 | 5.70 |
| Dermatology and Venereology | 10171 | 8.22 | 5.87 |
| Rehabilitation Medicine | 7806 | 8.10 | 5.99 |
| Anesthesiology | 514 | 8.24 | 6.36 |
| Traditional Chinese Medicine (TCM) | 2869 | 8.00 | 6.26 |
| Internal Medicine | 78665 | 8.36 | 6.36 |
| Surgery | 23541 | 8.32 | 6.37 |
| Obstetrics and Gynecology | 21227 | 8.33 | 6.31 |
| Pediatrics | 14253 | 8.26 | 6.26 |
| Otorhinolaryngology | 8519 | 8.28 | 5.96 |
| Other Departments | 158629 | 8.07 | 6.16 |
| Medical | 304858 | 8.20 | 6.24 |
| DPO Dataset | 32919 | 8.89 | 8.03 |

Table 17: Summary of key statistics and attributes of the FineMed dataset. **Quality** and **Complexity** refer to the average quality score and average complexity score, respectively. The table is divided into three sections: the first presents data from third-level departments, the second from second-level departments, and the third from first-level departments. Note that data with complexity score of 8 or higher have been extracted into the DPO dataset, so the average complexity score of the remaining data are comparatively lower.

## G   Case Study on the Application of DPO and TTT

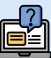 **A question from MMLU-Pro**

Question:
Polio can be eradicated by which of the following?

Options:
A. Herbal remedies
B. Use of antibiotics
C. Regular intake of vitamins
D. Administration of tetanus vaccine
E. Attention to sewage control and hygiene
F. Natural immunity acquired through exposure
G. Use of antiviral drugs

Ground-truth answer: E.

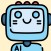 Answer from FineMedLM

To eradicate polio, it is essential to focus on public health measures that prevent the spread of the virus. This includes improving sanitation, ensuring access to clean water, and promoting hygiene practices such as handwashing. Additionally, vaccination campaigns are critical, as they provide immunity to the population and prevent the virus from spreading. The correct answer is E, which emphasizes the importance of sewage control and hygiene in preventing the transmission of polio.

🎓 Answer from FineMedLM-o1

**Thought**

So I have this question about polio and how it can be eradicated. There are several options presented, and I need to choose the one that's correct. First, I need to understand what polio is and how it's typically managed or prevented. (*planning*)

Polio, or poliomyelitis, is a highly contagious viral disease that can lead to paralysis and death. It's primarily spread through contaminated water or food, and it's caused by the poliovirus, which is a member of the Picornaviridae family. The disease has been a major public health concern, but over the years, vaccination programs have significantly reduced its incidence worldwide.

Looking at the options, I need to identify which ones are directly related to preventing or eradicating polio.

Option A: Herbal remedies. I'm not aware of any herbal remedies that can prevent or cure polio. Polio is caused by a virus, and viruses are not treated with herbs; antibiotics are ineffective against viruses. So, I can cross this one out.

Option B: Use of antibiotics. Antibiotics are used to treat bacterial infections, not viral infections like polio. Therefore, this is incorrect.

Option C: Regular intake of vitamins. Vitamins are important for overall health, but they don't prevent or treat viral infections. While a balanced diet rich in vitamins can support the immune system, it's not a direct method to eradicate polio.

Option D: Administration of tetanus vaccine. Tetanus is a bacterial infection caused by Clostridium tetani, which is unrelated to polio. The polio vaccine is specifically designed to prevent polio, so this is not the correct answer.

Option E: Attention to sewage control and hygiene. Polio is spread through contaminated water and food, so improving sanitation and hygiene practices can help prevent its transmission. This seems like a key factor in controlling the spread of the virus, so it might be part of the answer.

Option F: Natural immunity acquired through exposure. This is a controversial approach, and it's generally not recommended because the risks of paralysis and death from the disease far outweigh any potential benefits of natural immunity. Additionally, not everyone exposed to the virus will become infected, so relying on natural immunity is not a reliable strategy for eradication.

Option G: Use of antiviral drugs. There are antiviral drugs available for some viral infections, but polio is one of the viruses for which there are no specific antiviral treatments. The polio vaccine is the primary preventive measure.

Given that the polio vaccine is the most effective tool for preventing polio, and that sewage control and hygiene play a crucial role in preventing transmission, I think the correct answer is a combination of both.

*analysis*

**Summarization**

Polio can be eradicated through a combination of vaccination and improving hygiene and sanitation practices. The polio vaccine is the primary preventive measure, providing immunity against the virus. Additionally, ensuring proper sewage control and maintaining hygiene helps prevent the spread of the virus through contaminated water and food. Therefore, the correct answer is E, which emphasizes the importance of attention to sewage control and hygiene in the eradication of polio.

**Answer from FineMedLM-o1 with TTT (more thinking behavior)**

**Thought**

So I've got this question about polio and I need to figure out which of these options can help eradicate it. First, I need to understand what polio is and how it's transmitted. (*planning*) From the text, it mentions that polio is an acute infectious disease caused by a virus and that it can lead to paralysis. It's spread through contaminated food and water, and it's most common in tropical and subtropical regions where sanitation is poor.

Okay, so sanitation and hygiene seem really important here. The text mentions that sewage control is crucial, so that probably means maintaining clean water and sanitation facilities to prevent the spread of the virus. So, maybe option E is right because attention to sewage control and hygiene can help prevent the spread of the virus.    *analysis*

Also, it says that the virus is more common in areas where sanitation is poor, so improving sanitation could reduce the spread of polio. That makes sense.

Now, looking at the other options: (*exploration*)

A. Herbal remedies: The text doesn't mention anything about herbal remedies for polio. It talks about using herbal remedies for other conditions like constipation and diarrhea, but not for polio. So, probably not.

B. Use of antibiotics: Polio is caused by a virus, not bacteria, so antibiotics wouldn't help. Antibiotics are for bacterial infections, not viral ones.

C. Regular intake of vitamins: Vitamins can be important for overall health, but they don't treat viral infections like polio. So, probably not.

D. Administration of tetanus vaccine: Tetanus is caused by bacteria, not viruses, so it's unrelated to polio. The text mentions that polio is caused by a virus, so this probably isn't the answer.

E. Attention to sewage control and hygiene: This seems directly related to preventing the spread of polio, so it might be the answer.    *analysis*

F. Natural immunity acquired through exposure: The text says that polio is most common in areas where sanitation is poor, and it's an acute infectious disease caused by a virus. It's not something you want to acquire naturally because it can lead to paralysis. So, probably not.

G. Use of antiviral drugs: The text doesn't mention antiviral drugs for polio, and since it's a viral infection, antivirals might be helpful, but I don't know for sure. Maybe.

Wait (*reflection*) , the text says that polio can be eradicated by controlling sewage and maintaining sanitation. So, it's more about prevention than treatment. That suggests that keeping the environment clean prevents the virus from spreading, which makes sense.

So, based on that, option E seems to be the right answer.

**Summarization**

Polio can be eradicated primarily through attention to sewage control and hygiene. This is because poor sanitation contributes to the spread of the poliovirus, and maintaining clean water and sanitation facilities reduces the risk of infection. Therefore, the answer is E.

## H   Comparative Results on Internal Medicine and Endocrinology Benchmarks

| Model | Internal Medicine | Endocrinology |
|---|---|---|
| HuatuoGPT2-7B(Chen et al., 2023) | 54.77 | 42.92 |
| Medical-Llama3-8B | 42.33 | 40.17 |
| Llama3.1-8B(Dubey et al., 2024) | 40.77 | 38.99 |
| FineMedLM-s1 | 55.01 | 43.25 |
| FineMedLM-s2 | 60.55 | 46.33 |
| FineMedLM | **62.50** | **50.84** |

Table 18: Main results on the internal medicine and endocrinology benchmarks. These benchmarks are constructed from private datasets collected within the hospital. FineMedLM-s1 and FineMedLM-s2 refer to models fine-tuned in the first and second stages, respectively.

