# OpenReview forum: "FineMedLM-o1: Enhancing Medical Knowledge Reasoning Ability of LLM from Supervised Fine-Tuning to Test-Time Training"
_colmweb.org/COLM/2025/Conference — COLM 2025_

### Official Review · Reviewer_Xx4M · 2025-05-04

**Rating:** 5
**Confidence:** 3
**Ethics Flag:** 1

**Summary:**

This paper describes a pipeline for creating a medicine-specific reasoning LLM, from dataset creation, to fine-tuning steps, to test-time training and reasoning outputs. On benchmark datasets their model outperforms other open medical LLMs, and being based on Llama-8B, is competitive with GPT-4o-mini. It is not competitive with larger GPT models, QwQ, or DeepSeek-v3.
In my reading the two insights that provide the most leverage are curation of high quality medical data, and distillation of Qwen and QwQ, which they use for generating questions and reasonning responses for DPO training, respectively.

Their dataset creation pipeline starts with a highly classified dataset, FineFineWeb, from which they take the medical subset. They use Qwen to generate pairs of instructions (i.e. questions?) and rate them on quality, complexity, and medical relevance, and select one based on a rule-based combination of those scores.

They then generate responses, using Qwen for "common" questions and QwQ to respond to "complex" questions with long-form reasoning.

They also classify their data hierarchically by topic, essentially primary and secondary specialties, which they use later for curriculumn learning.

Their training process starts from Llama3.1-8B, sequentially fine-tune on a sample of all medical data, then primary specialty data, then secondary specialty data. This model they call FineMedLM. I infer that there are actually 29 of these since there are 29 secondary specialties? They give the example of internal medicine->endocrinology, but I'm assuming that's just a stand-in for any primary/secondary specialty pair they could train on.

Next they train for reasoning, by taking FineMedLM and first conducting SFT with reasoning data, and then a preference learning stage. This model they call FineMedLM-o1.

Finally, they also perform Test-Time Training (TTT). I wasn't familiar with this technique, but it is a retrieval step that gets one similar instance, followed by training on that instance, followed by generation on the benchmark instance. The model trained for that instance is discarded and subsequent instances are answered starting from FineMedLM-o1.

They evaluate on Chinese and English benchmark datasets. Their model out-performs other open models of similar size significantly, including those specific to medicine. On MMLU-Pro they compare to HuatuoGPT-o1, as well as GPT-4o-mini, QwQ-32B-preview, GPT-4o, and DeepSeek-v3. They don't compare to OpenAI reasoning models. They also only compare against HuatuoGPT-o1 for the second evaluation, and not the first. This seems like a somewhat unfair setup, since they show all of their models (with reasoning, with TTT) for that evaluation. They compare against DeepSeek-v3 but not the reasoning model. Overall, trying to extract a fair comparison from Table 1 ("Common medical benchmarks"), their LM is better on the English datasets but worse on the Chinese than Baichuan2-7B.
For Table 2 ("Complex reasoning benchmarks"), their model is similar to HuatuoGPT-o1 without test-time reasoning, but better with it.  The larger models are better (GPT-4o, DeepSeek-v3), even without using the reasoning models.

Overall, I think the most interesting thing here is the pipeline to create the dataset, and the fact that good 8B models can be obtained, using mostly synthetic data generation, distillation from larger open models, and lowering GPU requirements (4xA100 is not nothing but seems like progress). This provides some hope that highly capable domain specific open models may be obtainable, which would be great for democratization of these tools for study and non-corporate usage. However, the paper focuses on performance numbers, which show the model is about as good as GPT-4o-mini, which is dirt cheap and obviously much easier than creating one's own model. The other thing is, this kind of paper that does an end-to-end setup, while being impressive, is never satisfying as a scientific work because there are too many questions in such a complex pipeline to evaluate them all thoroughly.

**Reasons To Accept:**

* Interesting pipeline for creating reasoning datasets
* Potentially lowering resource requirements for training reasoning models
* Impressive end-to-end work

**Reasons To Reject:**

* Ultimately performance tops out near the performance of a really cheap proprietary model that isn't even trained for reasoning
* End-to-end paper writing means there are many research questions/decisions that are skipped over that may have major impact

---

> ### Author Response · Authors · 2025-06-03
>
> Thanks for your valuable questions! Below, we provide some responses and hope to address your concerns.
>
> **Answer to R1: Experimental Results Explanation**
>
> Thank you for the comment. In our experiments, we primarily compare our model with open-source general-purpose and medical-domain models of similar size, focusing on settings with limited computational resources. While proprietary models such as GPT-4o-mini and DeepSeek-V3 benefit from significantly larger training budgets and data, we also include their results as a reference.
> Importantly, our comparisons emphasize models like HuatuoGPT-o1, which are specifically trained under constrained resources and demonstrate strong reasoning capabilities in the medical domain. We believe these models provide a fairer and more relevant benchmark for evaluating our approach in realistic, resource-limited scenarios.
>
> **Answer to R2: key Research Questions and Decisions Explanation**
>
> Thank you for your comment. We respectfully argue with the suggestion that our paper skips over key research questions or decisions. On the contrary, our work explicitly identifies and addresses the main challenges in building a high-quality medical LLM system. Below, we summarize the core research questions and decisions, all of which are clearly presented and supported by detailed analysis in the paper.
>
> **Key research questions addressed:**
>
> 1. **Effectiveness of synthetic data generation.** Section 2.3 analyzes the semantic distribution of data across different first-level departments within FineMed, confirming data diversity and validating our low-cost, automatic classification method. We further benchmark our data quality using both LLM-as-a-judge and doctor evaluations, demonstrating that our synthetic data outperforms strong existing datasets.
>
> 2. **Effectiveness of three-stage SFT for medical knowledge learning.** Section 4.3 presents ablation studies across multiple benchmarks, comparing against single-stage training and reversed stage order. Performance on additional private internal medicine and endocrinology benchmarks shows the effectiveness of our design.
>
> 3. **Impact of TTT on medical reasoning.** Sections 4.2 and 4.3 demonstrate effects of TTT via performance improvements, ablations using different data types, and case study showing more advanced reasoning behavior, such as exploration and self-reflection.
>
> **Key design decisions explained:**
>
> 1. **Synthetic data pipeline.** Section 2.1 describes the full process. Appendices A–D further include prompts, scoring algorithms, and medical taxonomy details.
>
> 2. **Post-training and TTT configurations.** Sections 3.1–3.3 provide comprehensive training details, including computing resources, data volumes, and hyperparameters.
>
> Finally, all code and data generation processes are open-sourced to ensure transparency and reproducibility.
>
> We hope this clarifies that our work does not skip over key elements but rather addresses them thoroughly with extensive evidence.

---

> > ### Comment · Reviewer_Xx4M · 2025-06-06
> >
> > I have read and considered your response, and will keep my review as it is.

---

> > > ### Author Response · Authors · 2025-06-07
> > >
> > > Thank you sincerely for your detailed and helpful review! We hope our response have effectively addressed your concerns and would greatly appreciate the opportunity to further integrate your suggestions to enhance our work. Please let us know if you have any additional questions or comments.

---

### Official Review · Reviewer_H2Gw · 2025-05-06

**Rating:** 6
**Confidence:** 5
**Ethics Flag:** 1

**Summary:**

The paper proposes a dataset for both SFT and DPO in medical domain. Using medical-related texts from FineFineWeb, all the instructions and responses are generated by prompting Qwen and QwQ, and undergo a filtering process. The authors then propose a two-step training procedure (first a 3-stage SFT, then SFT with reasoning+DPO) to let the model progressively gain reasoning ability. Ablation experiments show effectiveness of this training pipeline, and the resulted model outperform baselines on multiple English and Chinese benchmarks.

**Questions To Authors:**

1. Texts in Fig1 are almost unreadable unless zooming in for 340%. It’s better to keep the key texts only.

2. There lacks detail of the data filtering algorithm (algo1) for the generated instructions. For example, what is “MentionSpecificDetails” in algorithm1?

3. In line 107-109, the authors mention “To ensure the instructions remain relevant to medicine and do not excessively dilute the total quality and complexity scores, relevance to medicine is scored on a scale of 1 to 6.” Can you explain more on the motivation?

4. **No answer verification is applied to the response generation stage**. Answer correctness should supersede all other metrics in the medical domain. However, no expert verification is conducted which poses a considerable concern on the data correctness. Benchmark performance isn’t everything. This would be the major issue to me for this paper to be accepted.

5. It’s better to summarize the key info and stats of the proposed dataset in a table.

6. Despite showing decent results, would you also show the increased inference cost introduced by TTT? It’s crucial to balance efficiency and efficacy, albeit the latter is more important in medicine.

7. It seems that the DPO dataset labels reasoning answers as positive and non-reasoning answers as negative. If this is the case, the potential of DPO may not be fully leveraged due to the large distribution gap between the paired samples. Do you think it’s better to construct paired answers using reasoning data only, with one of the samples deliberately made wrong?

**Reasons To Accept:**

1. A two-step training approach is proposed to enable the model to progressively gain reasoning ability.

2. Applying test time training to the medical domain seems beneficial.

3. Both training approaches are validated via ablations, which could bring insights to the community.

**Reasons To Reject:**

See below.

---

> ### Author Response · Authors · 2025-06-03
>
> Thank you for your valuable feedback! Below, we provide some responses and hope to address your concerns.
>
> **Answer to Q1: Figure Modification**
>
> Thank you for your suggestion. We have revised Figure 1 by enlarging the key text elements and removing non-essential details to improve readability without requiring excessive zooming.
>
> **Answer to Q2: Algorithm Details Providation**
>
> Thank you for your suggestion. We have clarified Algorithm 1 in the appendix by adding more detailed descriptions of its input and scoring mechanism. Specifically, each instruction is associated with a score set that includes three evaluation metrics—quality, complexity, and RelevanceToMedicine—along with a boolean indicator, MentionSpecificDetails, which denotes whether the response includes specific details.
>
> **Answer to Q3: Scoring Motivation Explanation**
>
> Thank you for the question. INSTAG [1] demonstrates that the overall quality and complexity of the SFT dataset are primarily influenced by the characteristics of the instructions. So we assign medical relevance a smaller range (1–6) than quality and complexity (each 1–10) to ensure that the overall score remains primarily driven by quality and complexity. This design reflects our intent to emphasize instruction quality and complexity, while still accounting for medical relevance without letting it disproportionately influence the total score.
>
> **Answer to Q4: Response Verification**
>
> Thank you for the insightful comment. Our methodology is designed to ensure answer correctness by construction: since both the instructions and responses are derived from source texts, and the model is constrained to retrieve and respond strictly based on this content, factual consistency is inherently preserved. Nonetheless, given the critical importance of correctness in the medical domain, we further strengthen our pipeline with an additional verification step. Specifically, an LLM-based verifier assesses instruction compliance and faithfulness to the seed text. Responses flagged as potentially problematic are subsequently reviewed and, if needed, refined by licensed physicians to ensure medical accuracy.
>
> **Answer to Q5: Key Information Summarization**
>
> Thank you for your valuable suggestion. We have summarized the key dataset statistics in a new table (added to Appendix E). A preview is included below for convenience. Note that the first row shows lower complexity score as samples with scores ≥8 are allocated to the DPO dataset.
>
> | Medical Specialty                        | Count  | Quality | Complexity |
> |------------------------------------------|--------|---------|------------|
> | Endocrinology                            | 11477  | 8.39    | 6.43       |
> | DPO Dataset                              | 32919  | 8.89    | 8.03       |
>
> **Answer to Q6: TTT Latency Comparison**
>
> Thank you for the suggestion. Below we report the average inference time table. The average inference time per instance increases by ~4.7× when moving from FineMedLM to FineMedLM-o1, due to the added intermediate thinking steps. Applying TTT further adds ~2.3× overhead. Despite this increase, the model produces notably more coherent and effective multi-step reasoning.
>
> We will include this latency comparison and analysis in the revised version to help better quantify the tradeoff between inference cost and reasoning performance.
>
> | Model                          | Inference Time | Thinking Behavior |
> |--------------------------------|----------------|--------------------|
> | FineMedLM                      | 3.3s           | Absent             |
> | FineMedLM-o1                   | 15.4s          | Present            |
> | FineMedLM-o1 with TTT          | 35.1s          | Enhanced           |
>
> **Answer to Q7: DPO Dataset Construction**
>
> Thank you for your insightful question and suggestion regarding our DPO dataset construction.
>
> In our current setup, we intentionally include non-reasoning responses as negative samples. This design aims to explicitly teach the model the *output pattern* associated with long-CoT reasoning by contrasting it against outputs lacking this reasoning structure.
>
> We agree that exploring the pairing strategy you propose (using only reasoning data, with one deliberately incorrect sample) is valuable. To investigate this, we plan the following:
>
> 1. Generate responses using FineMedLM.
>
> 2. Verify the correctness of these responses.
>
> 3. Select incorrect responses (regardless of whether they contain reasoning or not) as negative samples to pair with correct reasoning responses.
>
> We believe this targeted approach of learning from model-generated errors within the reasoning paradigm will allow the model to better refine its reasoning capabilities based on its current deficiencies.
>
> Thanks for these questions. We will add these details to the paper.
>
> [1] Lu, Keming, et al. "# instag: Instruction tagging for analyzing supervised fine-tuning of large language models." arXiv preprint arXiv:2308.07074 (2023).

---

> > ### Comment · Reviewer_H2Gw · 2025-06-07
> > **Thanks for your reply!**
> >
> > Thank you for your detailed reply. I've raised my score and confidence level.

---

> > > ### Author Response · Authors · 2025-06-07
> > >
> > > Thank you again for reviewing our work! We are glad that our response addressed the main concerns, and we greatly appreciate your detailed feedback and suggestions throughout the review process.

---

### Official Review · Reviewer_NP49 · 2025-05-13

**Rating:** 7
**Confidence:** 5
**Ethics Flag:** 1

**Summary:**

The paper proposes FineMedLM-o1, a medical reasoning model trained from synthetic data, SFT, DPO, and TTT. The paper is well written. The training recipe is quite standard. However, the data quality of the generated dataset is not clear enough and some baseline methods are missing. The analysis can be more thorough than pure quantitative metrics. The latency of TTT can be helpful. Overall, the paper can be valuable for medical AI domain and inspire future work in medical reasoning models.

**Reasons To Accept:**

1. The writing is very clear. The training recipe is standard and can inspire other future work.
2. The evaluation on multiple benchmarks is comprehensive, and demonstrates the good performance of the model.

**Reasons To Reject:**

1. It seems that there is no filtering when generating the synthetic training data, so the data quality may not be guaranteed. Hallucinations can be very harmful, particularly in the medical domain.
2. There's no qualitative analysis on the model performance. For example, a case study about how the DPO and TTT work can be helpful.
3. There are previous work that used similar training recipe to train medical LLMs, e.g. UltraMedical [1]. The authors are expected to compare and discuss the differences.
4. The TTT method used in this paper requires retrieval and further training for every each instance during testing. This seems time-consuming. The authors are expected to report and compare the latency with and without TTT.

[1] Zhang, Kaiyan, et al. "Ultramedical: Building specialized generalists in biomedicine." Advances in Neural Information Processing Systems 37 (2024): 26045-26081.

---

> ### Author Response · Authors · 2025-06-03
>
> Thank you for your valuable feedback! Below, we provide some responses and hope to address your concerns.
>
> **Answer to R1: Synthetic Data Quality Filtering**
>
> We appreciate your concern regarding data quality and potential hallucinations in synthetic training data. Our data generation pipeline is built on seeds from FineFineWeb [1], which in turn are derived from FineWeb [2]. Both sources apply multiple rounds of deduplication and quality filtering to ensure the diversity and reliability of the base data.
>
> During instruction-response synthesis, we extract instructions from these high-quality seeds and generate responses by prompting the LLM with both the instruction and its corresponding seed document. This grounding ensures that the generated answers are traceable to original, verified content, thereby reducing the risk of hallucinations.
>
> In response to your suggestion, we have further enhanced our pipeline by introducing an additional verification step. Specifically, we use an LLM-based verifier to (1) check whether the response appropriately follows the instruction, and (2) determine whether the content is faithful to the original seed. For cases flagged as potentially problematic, professional physicians manually review and, if necessary, revise the responses to ensure medical correctness and completeness.
>
> **Answer to R2: Case Study**
>
> Thank you for the helpful suggestion. In response, we have added a case study in the paper to qualitatively compare the outputs of FineMedLM (before DPO), FineMedLM-o1 (after DPO), and FineMedLM-o1 with TTT. The example is selected from the MMLU-Pro benchmark and involves a complex reasoning task.
>
> The comparison illustrates how DPO enhances the model's ability to exhibit structured reasoning behaviors such as planning and analytical thinking. Moreover, with TTT, the model demonstrates additional reasoning patterns that were not previously observed, including exploration and self-reflection. This analysis provides insights into how each stage of alignment contributes to the model’s reasoning capabilities.
>
> **Answer to R3: Differences Comparison**
>
> Thank you for the helpful suggestion. The frameworks of all synthetic data methods are very similar. Our motivation is to give the model strong medical reasoning capabilities, so we differ from UltraMedical [3] in terms of the focus of data generation and the capabilities of the final model.
>
> 1. **Focus of Synthetic Data**: Our work emphasizes the generation of high-quality, high-complexity instructions paired with reasoning-intensive responses, including both direct answers and long-form reasoning processes. This design aims to explicitly strengthen the model’s **reasoning and reflective thinking abilities**. UltraMedical focuses on maximizing the quality and diversity of data, and obtains models with rich biomedical knowledge. However, UltraMedical does not consider the effect of long-form reasoning data on improving knowledge manipulation capabilities of the model.
>
> 2. **Capabilities of the Final Model**: Our model is optimized for **robust reasoning** and exhibits emergent behaviors in multi-step reasoning and knowledge retrieval tasks, especially after applying TTT. In contrast, UltraMedical primarily enhances **knowledge retrieval and factual accuracy** through multi-step optimization strategy, but has not been tested on reasoning tasks. Thus, although both methods leverage synthetic data, they target **distinct downstream competencies**.
>
> **Answer to R4: TTT Latency Comparison**
>
> Thank you for the suggestion. Below we report the average inference time table. The average inference time per instance increases by ~4.7× when moving from FineMedLM to FineMedLM-o1, due to the added intermediate thinking steps. Applying TTT further adds ~2.3× overhead. Despite this increase, the model produces notably more coherent and effective multi-step reasoning.
>
> We will include this latency comparison and analysis in the revised version to help better quantify the tradeoff between inference cost and reasoning performance.
>
> | Model                          | Inference Time | Thinking Behavior |
> |--------------------------------|----------------|--------------------|
> | FineMedLM                      | 3.3s           | Absent             |
> | FineMedLM-o1                   | 15.4s          | Present            |
> | FineMedLM-o1 with TTT          | 35.1s          | Enhanced           |
>
> Thanks for these questions. We will add these details to the paper.
>
> [1] https://huggingface.co/datasets/m-a-p/FineFineWeb
>
> [2] Penedo, Guilherme, et al. "The fineweb datasets: Decanting the web for the finest text data at scale." Advances in Neural Information Processing Systems 37 (2024): 30811-30849.
>
> [3] Zhang, Kaiyan, et al. "Ultramedical: Building specialized generalists in biomedicine." Advances in Neural Information Processing Systems 37 (2024): 26045-26081.

---

> > ### Comment · Reviewer_NP49 · 2025-06-06
> >
> > Thank you for the detailed reply. My concerns are well addressed and I increase my score from 6 to 7.

---

> > ### Author Response · Authors · 2025-06-06
> >
> > Thank you again for reviewing our work! We are pleased that our reply addressed the main concerns. Your detailed feedback and suggestions throughout the entire process have been invaluable, and we will incorporate them into our updated manuscript.

---

### Official Review · Reviewer_BAG1 · 2025-05-13

**Rating:** 7
**Confidence:** 4
**Ethics Flag:** 1

**Summary:**

The paper introduces FineMedLM-o1, an advanced medical large language model (LLM) designed specifically to improve reasoning capabilities for complex medical tasks such as differential diagnosis and medication recommendations. To overcome existing LLM limitations in deep medical reasoning, the authors propose a three-stage training strategy using carefully synthesized medical dialogue data, including comprehensive long-form reasoning content (o1-style). The model training leverages Supervised Fine-Tuning (SFT), Direct Preference Optimization (DPO), and notably introduces Test-Time Training (TTT) into medical LLMs to enhance domain adaptation and reasoning performance. The model and training strategies are extensively evaluated on standard and challenging medical benchmarks, demonstrating significant performance improvements over previous models. FineMedLM-o1 outperforms several baseline medical models, with further performance gains observed upon incorporating TTT.

**Questions To Authors:**

See above.

**Reasons To Accept:**

- The authors introduce a framework to synthesize large-scale, high-quality medical dialogue data specifically tailored for medical LLM training. The dataset, called FineMed, comprises approximately 300,000 curated samples classified into fine-grained categories based on medical specialization, thus addressing limitations of previous medical datasets in quality and complexity. This could be a helpful resources for medical reasoning model development upon release.
- For the first time in medical LLMs, TTT is employed during inference, allowing the model to adapt dynamically to relevant medical knowledge for each query, substantially boosting reasoning performance, particularly on challenging benchmarks.
- Extensive evaluations demonstrate an average performance improvement of about 23% over prior models on key medical benchmarks, with an additional 14% improvement from TTT. The authors commit to releasing all code, datasets, and resources publicly to facilitate future research and innovation.

**Reasons To Reject:**

- My main concern comes from the data recipe in the three-stage SFT. The current ablation study seems not enough to directly demonstrate the effectiveness of each data cohort under three stages.
- Continue with my previous questions, the second and third stage goes to more domain-specific data including Internal Medicine and Endocrinology. From the evaluation set, I am not sure if there is a specific evaluation benchmark to evaluate Internal Medicine or Endocrinology specific reasoning capabilities.
- One of the major contributions to me is the high-quality SFT dataset annotated by 5+ experts. Please conduct more detailed quality check or evaluation to demonstrate the quality of the proposed dataset.

---

> ### Author Response · Authors · 2025-06-03
>
> Thanks for your valuable questions! Below, we provide some responses and hope to address your concerns.
>
> **Answer to R1&R2: 3-stage SFT Evaluations**
>
> As fine-grained medical datasets in subfields such as internal medicine and endocrinology remain scarce in the open-source community, we rely on a general-purpose medical benchmark for evaluation. This follows the precedent set by prior work such as PediatricsGPT [1], where the authors train a pediatric-specialized language model but also evaluate it on general medical benchmarks due to similar data limitations.
>
> To better illustrate the effect of the 3-stage SFT method and data, we conduct additional experiments using a private benchmark from a hospital dataset that specifically covers internal medicine and endocrinology. We evaluate the models trained at each stage of the proposed 3-stage SFT process (FineMedLM-s1, FineMedLM-s2, and FineMedLM), as well as several strong open-source baselines. The results are summarized in the table below:
>
> | Model                          | Internal Medicine | Endocrinology |
> |--------------------------------|--------------------|----------------|
> | HuatuoGPT2-7B| 54.77              | 42.92          |
> | Medical-Llama3-8B              | 42.33              | 40.17          |
> | Llama3.1-8B | 40.77              | 38.99          |
> | FineMedLM-s1                   | 55.01              | 43.25          |
> | FineMedLM-s2                   | 60.55              | 46.33          |
> | FineMedLM                      | 62.50              | 50.84          |
>
> **Answer to R3: Expert Quality Evaluation**
>
> Following the practice in Aquila-Med [2], we initially employ an LLM-based automatic evaluation to assess the quality and complexity of our dataset. We appreciate your suggestion that a more rigorous quality assessment would benefit the development of medical LLMs.
>
> To this end, we randomly sample 3,000 instances from the FineMed dataset and invite professional physicians to evaluate them using the same criteria as in the LLM-based assessment. The results are shown in the table below. For comparison, we also include the original LLM-based scores. Notably, we observe a strong alignment between the human and LLM evaluations, suggesting the reliability of our automatic assessment approach.
>
> | Evaluator        | Quality (Average Score) | Complexity (Average Score) |
> |------------------|-------------------------|----------------------------|
> | Expert           | 8.35                    | 6.33                       |
> | LLM              | 8.27                    | 6.41                       |
>
> Thanks for these questions. We will add these details to the paper.
>
> [1] Yang, Dingkang, et al. "Pediatricsgpt: Large language models as chinese medical assistants for pediatric applications." Advances in Neural Information Processing Systems 37 (2024): 138632-138662.
>
> [2] Zhao, Lulu, et al. "Aqulia-med llm: Pioneering full-process open-source medical language models." arXiv preprint arXiv:2406.12182 (2024).

---

> > ### Comment · Reviewer_BAG1 · 2025-06-03
> >
> > Thank you for your response. My previous concerns have been addressed.

---

> > ### Author Response · Authors · 2025-06-06
> >
> > Thank you again for reviewing our work! We are pleased that our response addressed the main concerns and sincerely appreciate your detailed feedback and suggestions throughout the review process.

---

### Decision · Program_Chairs · 2025-07-08

**Decision:**

Accept

**Comment:**

The paper presents FineMedLM-o1, a medical reasoning model developed through a structured training pipeline that integrates synthetic data generation, multi-stage supervised fine-tuning, Direct Preference Optimization (DPO), and test-time training (TTT). The reviewers generally recognize the merit of the proposed data curation strategy, the novel incorporation of TTT into medical LLMs, and the strong empirical results demonstrating consistent improvements on established benchmarks.

While Reviewer Xx4M remains unconvinced, citing concerns regarding the incremental nature of the contributions and the overall complexity of the pipeline, the remaining reviewers are in favor of acceptance, particularly in light of the authors’ detailed rebuttal, which effectively addresses issues related to data quality, ablation studies, latency, and evaluation methodology. For the final version, the authors are encouraged to (i) more explicitly discuss the individual contributions of each training stage, (ii) expand the qualitative analysis to include representative failure cases, (iii) clearly articulate the computational trade-offs associated with TTT, and (iv) improve the visual clarity of Figure 1 as well as the presentation of dataset statistics.